# Mitigating the Impact of Noise on Nonlinear Frequency Division Multiplexing †

**Stella Civelli** [1,2,*] , **Enrico Forestieri** [1,2] and **Marco Secondini** [1,2]

1   Tecip Institute, Scuola Superiore Sant'Anna, 56127 Pisa, Italy; e.forestieri@santannapisa.it (E.F.);
    m.secondini@santannapisa.it (M.S.)
2   PNTLab, Consorzio Nazionale Interuniversitario per le Telecomunicazioni (CNIT), 56127 Pisa, Italy
*   Correspondence: s.civelli@santannapisa.it
†   This paper is an extended version of our paper published in 2020 Optical Fiber Communications Conference
    and Exhibition (OFC).

**Abstract:** In the past years, nonlinear frequency division multiplexing (NFDM) has been investigated as a potentially revolutionary technique for nonlinear optical fiber communication. However, while NFDM is able to exploit the Kerr nonlinearity, its performance lags behind that of conventional systems. In this work, we first highlight that current implementations of NFDM are strongly suboptimal, and, consequently, oversensitive to noise: the modulation does not ensure a large minimum distance between waveforms, while the detection is not tailored to the statistics of noise. Next, we discuss improved detections strategies and modulation techniques, proposing some effective approaches able to improve NFDM. Different flavors of NFDM are compared through simulations, showing that (i) the NFDM performance can be significantly improved by employing more effective detection strategies, with a 5.6 dB gain in Q-factor obtained with the best strategy compared to the standard strategy; (ii) an additional gain of 2.7 dB is obtained by means of a simple power-tilt modulation strategy, bringing the total gain with respect to standard NFDM to 8.3 dB; and (iii) under some parameters range (rate efficiency $\eta \leq 30\%$), the combination of improved modulation and detection allows NFDM to outperform conventional systems using electronic dispersion compensation.

**Keywords:** nonlinear Fourier transform; nonlinear frequency division multiplexing; inverse scattering transform; Kerr nonlinearity; coherent communication

## 1. Introduction

Fiber nonlinearity can be regarded not only as a detrimental effect, but also as an opportunity for all-optical signal processing and regeneration and for the establishment of favorable conditions for signal propagation. Key properties of the fiber propagation equation are its integrability via the *nonlinear Fourier transform* (NFT) and the existence of soliton solutions [1]. In the context of fiber communications, these properties have fostered the dream of a sort of "transparent" propagation regime, virtually immune to both dispersion and nonlinearity. Soliton systems, where information is carried by solitary waves propagating unchanged through the fiber, have been deeply studied in the eighties and nineties of the last century [2]. More recently, also thanks to great progress in DSP technology, the same dream has been revived and widened by the proposal of a *nonlinear frequency-division multiplexing* (NFDM) scheme, which, similarly to the well known orthogonal frequency-division multiplexing (OFDM) for linear channels, uses the NFT to encode information on the *nonlinear spectrum* of the optical signal, relying on its substantial invariance during propagation [3–5]. In recent years, numerous theoretical and numerical investigations and experimental

demonstrations of NFDM have been reported [6–18], though, possibly due to the pioneering nature of the topic, a clear advantage over conventional systems has yet to be demonstrated.

A fundamental difference of NFDM with respect to OFDM is that the NFT, unlike the ordinary Fourier transform, is not a unitary transformation, so that the statistics of the amplified spontaneous emission (ASE) noise, modeled as additive white Gaussian noise (AWGN) in time domain, are not preserved in the nonlinear frequency domain [3,8,19–21]. These modified statistics result in an unfavorable dependence of performance on the signal duration, which practically limits the achievable spectral efficiency [9]. A possible explanation of this negative result comes from the use of detection strategies borrowed from conventional communication systems, and optimized for the AWGN channel rather than for the actual noise statistics. Indeed, a number of *improved* detection strategies have been proposed to address this issue [11,13,22–25] but, unfortunately, without fully solving it. Another possibility is that the linear modulations (e.g., quadrature amplitude modulation (QAM)) that are employed to encode the information on the nonlinear spectrum, while assuring a certain *minimum Euclidean distance* between the waveforms in the nonlinear frequency domain, resulting in a too-small distance in the time domain, making the system oversensitive to the ASE noise.

In this paper, after briefly introducing NFDM systems, we discuss the noise problem in their earliest implementations, highlighting why both the standard modulation and demodulation steps are not optimal [26]. In the light of the above, we discuss some improved modulation and detection strategies, introducing for the first time a power-tilt strategy to mitigate some of the aforementioned issues. The various strategies are then compared in terms of performance through numerical simulations. While highlighting the significant improvements obtained by properly designing the NFDM system, we remark the unfavorable behavior with respect to conventional (i.e., with no NFT) systems, which causes a performance decay of NFDM systems at high spectral efficiency. The analysis is performed in a simple ideal scenario, assuming ideal distributed amplification along the fiber, modulating only the continuous spectrum of a single polarization, and favoring accuracy over complexity in the implementation of the NFT algorithms, so that numerical issues with the latter are avoided. The rationale behind these choices is then discussed, considering the expected impact of loss and dual-polarization modulation, the possible role of the discrete spectrum, and the extension to a network scenario.

The paper is organized as follows. Section 2 describes a standard NFDM system setup, while Section 3 discusses its suboptimality. Next, improved detection strategies and modulation techniques for NFDM are discussed in Section 4 and in Section 5, respectively. Section 6 compares the performance of different NFDM systems. Section 7 comments on some yet-to-be-investigated concepts related to NFDM systems. Finally, Section 8 draws the conclusions.

## 2. NFDM System Setup

The NFDM setup considered in this work is sketched in the upper part of Figure 1 [13,24] and assumes that only the continuous part of the nonlinear spectrum is modulated. The encoder generates a 16-QAM signal $s(t)$ encoding $N_b$ symbols with symbol rate $R_s = 1/T_s = 50\,\mathrm{GBd}$, and adds a guard time of $N_z$ null symbols to account for burst transmission and group velocity dispersion (GVD). A number of techniques to map the QAM signal $s(t)$ to the nonlinear continuous spectrum exist. In this work, unless otherwise specified, the nonlinear inverse synthesis (NIS) mapping—one of the first proposed techniques [6]—is considered. The nonlinear spectrum $\rho(\lambda)$ is obtained by

$$\rho(\lambda) = -S(\lambda/\pi) \tag{1}$$

where $S(f)$ is the ordinary Fourier transform of $s(t)$. Other possible modulation techniques are described in Section 5. According to the NFT theory, the channel effects are fully precompensated at the transmitter (TX) by multiplying the signal $\rho(\lambda)$ by the term $\exp(4j\lambda^2\mathcal{L})$, with $\mathcal{L}$ being the normalized link length. While channel compensation can equivalently be done at the TX or at the

receiver (RX) for standard detection (and even split between TX and RX [9]), it should be performed at the TX when using the improved detection strategies discussed in Section 4. After precompensation, the backward NFT (BNFT) is performed, the signal is reversed in time (time inversion does not affect the system performance but is considered here for the sake of simplicity), and the optical signal is obtained through a digital-to-analog converter (DAC) and launched into the fiber. At the RX, an analog-to-digital converter (ADC) provides the samples $\tilde{\mathbf{r}}$ of the received signal, from which the transmitted information is extracted by using a given detection strategy. The standard detection strategy—referred to as forward NFT (FNFT) detection—computes the FNFT of the received samples and obtains a noisy version of the scattering data $a(\lambda)$ and $b(\lambda)$. The nonlinear continuous spectrum—where the information was encoded—is obtained by $\rho(\lambda) = b(\lambda)/a(\lambda)$. However, noise (and, possibly, numerical inaccuracies) can make $|a(\lambda)|$ very close to zero, causing $\rho(\lambda)$ to diverge and the performance to drop. This effect can be partially mitigated with the simple expedient of replacing $|a(\lambda)|$ with the average of the values in the adjacent $\lambda$ when it is smaller than a certain threshold [27], here taken equal to $10^{-1}$. After the FNFT, the detection proceeds as in a conventional receiver: the nonlinear spectrum undergoes matched filtering and sampling, and the decisions are made according to a minimum Euclidean distance criterion, usually implemented by means of multiple thresholds on the in-phase and quadrature components. Other detection strategies are discussed in Section 4.

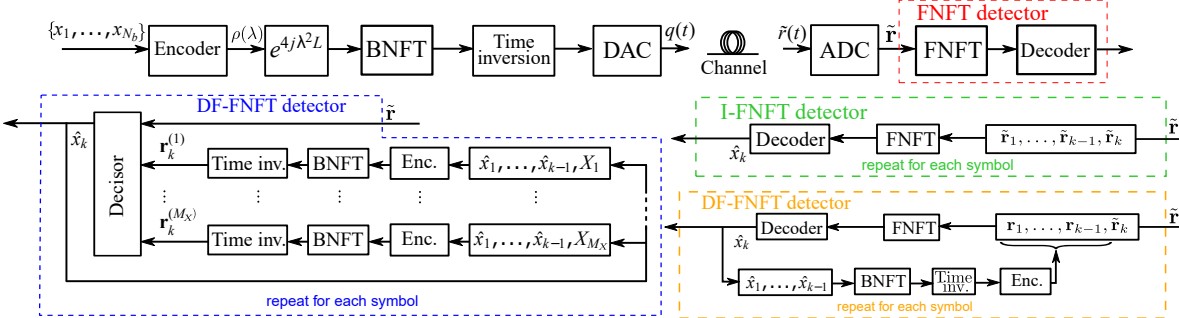

**Figure 1.** The nonlinear frequency-division multiplexing (NFDM) system with different detection strategies.

The physical channel is a single-mode fiber of length $L = 2000\,\text{km}$, attenuation $\alpha = 0.2\,\text{dB/km}$, GVD parameter $\beta_2 = -20.39\,\text{ps}^2/\text{km}$, and nonlinear coefficient $\gamma = 1.22\,\text{W}^{-1}\text{km}^{-1}$, along which ideal distributed amplification with spontaneous emission factor $\eta_{\text{sp}} = 4$ is considered. The bandwidth of both the DAC and the ADC is $100\,\text{GHz}$.

The system performance is reported in terms of $Q_{\text{dB}}^2$, where the $Q$-factor is evaluated from the bit error rate (BER), measured through direct error counting [13]. The performance is shown either as a function of the power $P = E_s/T_s$, with $E_s$ being the average symbol energy, or as a function of the rate efficiency $\eta = N_b/(N_b + N_z)$, which is changed by varying $N_b$ [9,13]. The number of null symbols is set to $N_z = 2000$ [13].

## 3. The Noise Problem

Conventional systems, i.e., systems using linear modulation with no NFT, are suitably optimized for the AWGN channel. On the one hand, the use of Nyquist pulses and a QAM format at the TX ensures that a certain minimum Euclidean distance between waveforms is maintained; this distance, which is *independent of the number of transmitted symbols* and *increases with the signal power*, determines the ultimate performance of the scheme. On the other hand, the maximum a posteriori probability (MAP) strategy is implemented at the RX by using matched filtering, sampling, and multiple thresholds to detect the closest symbol in Euclidean distance terms; this strategy ensures that the ultimate system performance is achieved. The same minimum Euclidean distance and the optimality of the detection strategy are preserved even in OFDM systems, since the IFFT and FFT operations, respectively performed at the TX and RX, are unitary transformations. Unfortunately, the fiber optic channel is not AWGN in

the nonlinear regime, bringing in additional issues, including signal-noise interaction and inter- and intra-channel nonlinearities.

In the recent years, NFT-based systems have been proposed to overcome the limitations due to fiber nonlinearity. Due to the pioneering and complex nature of the topic, these systems have been initially modeled along the lines of the conventional ones, with no guarantee of optimality. For example, NFDM was simply developed as a nonlinear analogous of OFDM, replacing IFFT with BNFT and FFT with FNFT. Unfortunately, the BNFT and the FNFT are not unitary transformations, so that the overall system is no longer optimal in terms of minimum distance, neither at the TX nor at the RX.

As far as it concerns the demodulation, the standard FNFT strategy [6] employs the FNFT followed by a conventional QAM detector, which is optimized for an AWGN channel. However, even in a back-to-back configuration, the ASE noise is AWGN in the time domain, but non-Gaussian and signal-dependent in the nonlinear frequency domain [8,9,19]. These modified statistics result in an unfavorable dependence of performance on the signal duration, which makes the FNFT strategy highly suboptimal, and practically limits the achievable spectral efficiency [9]. The importance of this issue is now widely acknowledged, so that some alternative detection strategies have been proposed [11,13,22–25].

On the other hand, also the modulation is affected by a similar issue, whose importance is, however, still largely overlooked. In fact, the direct mapping of the linearly modulated signal to the nonlinear spectrum warrants a certain minimum Euclidean distance between the waveforms in the nonlinear frequency domain but not in the time-domain, where the ASE noise is injected (ASE is typically the dominating noise in long-haul optical fiber systems, but the same issue applies to all the forms of noise that are injected between the BNFT and the FNFT, including for instance transceiver noise). If the distance between waveforms is too small in the time domain, the system becomes oversensitive to noise and the performance inevitably decays, even if an optimal strategy is employed. Unfortunately, this happens for the standard NIS modulation, as shown in Figure 2a, where the (upper bound to the) squared minimum distance among transmitted waveforms in time domain (after the BNFT) is shown as a function of the power for different burst lengths. The distance is normalized by twice the value of the standard deviation $\sigma$ of the noise generated in the considered link. For $N_b = 1$ the minimum distance monotonically increases with power (at least in the considered range), closely lower bounding the curve for linear modulation (valid for any $N_b$). On the other hand, for $N_b > 1$, the minimum distance reaches a maximum and then decreases—the longer the burst length $N_b$, the lower the peak value of the minimum distance.

The behavior of the minimum distance obtained with NFDM modulation is quite counterintuitive and totally different from that obtained with linear modulation, where, by contrast it grows monotonically with power and independently of the length of the transmitted sequence. This peculiar behavior highlights an intrinsic and critical limitation of NFDM modulation, which can in fact achieve a reasonable performance only with short bursts, i.e., for low spectral efficiencies, regardless of the adopted detection strategy.

Figure 2a reports the actual minimum distance only for the short bursts with $N_b = 1, 2, 3$. For longer bursts, since the exhaustive search of the minimum distance among all the possible waveforms, becomes too computationally expensive, the search was limited only to a subset of possible waveforms, hence obtaining an upper bound—in these cases, the actual minimum distance might indeed be even smaller, but not larger. An example of the upper bounding procedure for $N_b = 512$ is shown in Figure 2b, which reports the squared minimum distance found between waveforms when considering a fixed number of realizations and changing only some specific symbols. The figure reports, with the label "several symbols change", the squared minimum distance found among 150 random sequences. Furthermore, the figure shows the squared minimum distance found, among 150 random sequences, by changing just one symbol. This is done considering the sequences obtained by testing all the 16 constellation symbols in a fixed position: the change in the first position,

last position, and random position are indicated as "first symbol", "last symbol", and "random symbol", respectively. As expected, the distance is larger when changing a random number of symbols in a random position, and smaller when changing only one symbol. In particular, when the first symbol is changed, the minimum distance is small at low power and increases at high power, approaching the curve with several changes. Conversely, when only the last symbol is changed, the smallest minimum distance is obtained. In this case, the minimum distance reaches a maximum and then decreases. This happens because, according to the causality property of the NFT (see next section), the first symbol influences all the signal, while the last symbol only the tail, this effect being negligible at low power and strong at high power. According to this analysis, the best lower bound is obtained when changing only the last symbol of the sequence. This is, in fact, the bound considered in Figure 2a for $N_b > 3$. Finally, Figure 2b suggests that NFDM modulation induces an uneven behavior along the time-domain waveform—variations of the first symbols of the sequence cause larger changes of the waveform than variations of the last ones—revealing possible ways to improve the overall performance, as it will be shown in Section 5.3.

In conclusion, while a clear advantage of NFDM systems over conventional ones has not been yet demonstrated, NFDM can still be improved. Aiming at an optimal system, both the modulation and the detection steps should be modified and tailored for the NFDM paradigm. Possible improvements are discussed in the next sections.

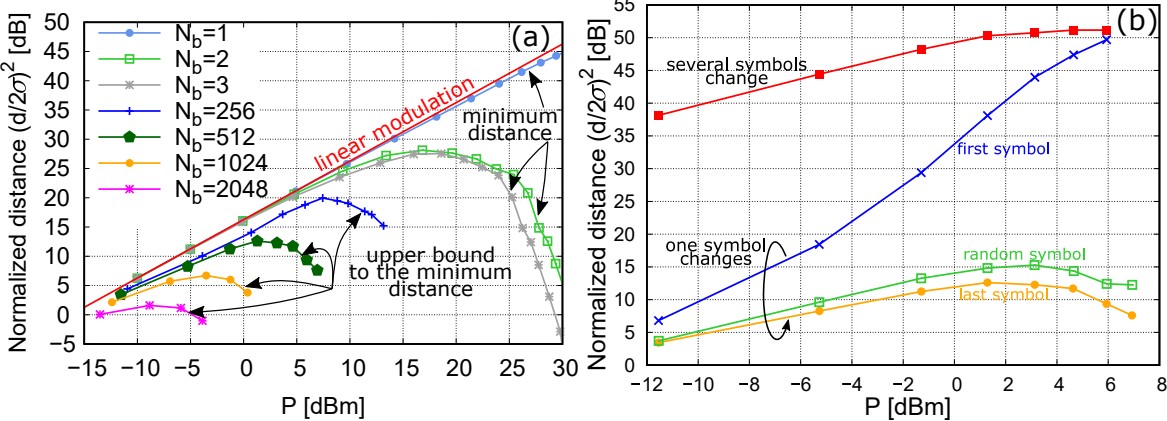

**Figure 2.** Nonlinear inverse synthesis (NIS) modulation (**a**) Normalized (upper bound to the) squared minimum distance between transmitted NFDM waveforms. (**b**) Normalized squared minimum distance obtained after several realizations changing a different set of symbols.

## 4. Improved Detections

Aiming at mitigating the noise problem and improving the performance of NFDM, some improved detection strategies have been proposed. For example, in the context of a discrete-spectrum modulation, the authors of [22] have proposed a time-domain strategy to detect a signal with seven eigenvalues, whose spectral amplitudes are modulated by QPSK symbols. The strategy consists of considering all the $2^{14}$ possible combinations of symbols and selecting the one that corresponds to the time-domain waveform that is closest to the received one in Euclidean sense. Unfortunately, the complexity of this optimal strategy increases exponentially with the number of eigenvalues and the constellation order. In a similar context, a machine learning technique has been also proposed for detection, providing superior performance with respect to the standard FNFT detection [25].

In principle, the optimal MAP strategy proposed in [22] for the detection of the discrete spectrum could be extended also to the scenario considered in this work for the detection of the continuous spectrum: compute the Euclidean distance between the received waveform and the $16^{N_b}$ waveforms corresponding to all the possible transmitted sequences of $N_b$ 16-QAM symbols, and select the sequence that minimizes this distance. However, this strategy is not feasible in practice, since it would require the computation (or storage) of $16^{N_b}$ signals and the computation of all the corresponding

distances. A possible way to circumvent this difficulty is to resort to the causality property of the NFT demonstrated in [13]. Indeed, the combination of this property with the sequential structure of some NFT algorithms allows us to develop some improved detection techniques which, though suboptimal, yield an improved performance compared to the standard FNFT detection without an exponential increase of complexity [13,23,24]. In the remainder of this section, we illustrate the NFT causality property and its implications in terms of detection, we describe three improved detection strategies, and we discuss their complexity. The notations are as in [13], and $\tilde{\mathbf{r}}_k$ (and $\mathbf{r}_k$) indicates the vector containing the samples of the noisy (and noiseless) optical signal in the time window $[t_{k-1}, t_k)$.

### 4.1. NFT Causality Property

The causality property of the NFT refers to the existence of a causality relation between the kernel of the Gelfand–Levitan–Marchenko equation, which is related to the nonlinear spectrum of the signal, and the corresponding time-domain signal that is obtained from the solution of this equation [13]. This property implies that if (i) the NIS modulation Equation (1) is used, (ii) full channel precompensation is employed, and (iii) the QAM signal $s(t)$ is obtained from a pulse shape with a duration smaller than the symbol time $T_s$, then the optical signal received before a certain time instant $t_k$ depends only on the first $k$ symbols $x_1, \dots, x_k$, and not on the following ones.

### 4.2. The Iterative-FNFT (IFNT) Detection

The incremental FNFT (I-FNFT) detection—sketched in Figure 1 in green—uses the NFT causality property to reduce the noise entering the standard FNFT detection, applying an iterative detection [24]. When detecting the $k$-th symbol, the I-FNFT strategy processes the signal samples up to the time instant $t_k$, such that the noise located after $t_k$ does not enter the FNFT, resulting in a reduction of noise in the nonlinear frequency domain, where symbols are detected. The pseudocode for the I-FNFT detection is shown in Algorithm 1.

---

**Algorithm 1** I-FNFT detection

---

1: **Data:** Samples of the received signal $\tilde{\mathbf{r}} = (\tilde{\mathbf{r}}_1, \dots, \tilde{\mathbf{r}}_{N_b})$
2: **Result:** The detected symbols $\hat{\mathbf{x}}_1, \dots, \hat{\mathbf{x}}_{N_b}$
3: **for** $k = 1, \dots, N_b$ **do**

4:     Apply the FNFT to $\tilde{\mathbf{r}}_1, \dots, \tilde{\mathbf{r}}_k$
5:     Obtain a noisy version of the nonlinear spectrum
6:     Apply standard detection (matched filtering and sampling)
7:     Obtain $\hat{x}_k$
8: **end for**

---

The computational cost of the I-FNFT is that of a single FNFT, since the nonlinear spectrum is evaluated independently on each nonlinear frequency, by using, for example, the Boffetta–Osborne method [28].

### 4.3. The Decision Feedback-FNFT (DF-FNFT) Detection

The decision-feedback FNFT (DF-FNFT) detection—sketched in Figure 1 in red—is an improvement of the I-FNFT. After applying the same procedure of the I-FNFT (iterative detection in the nonlinear frequency domain, removing part of the signal), the DF-FNFT cleans the signal from the noise before the $k$-th symbol, using the feedback from already detected symbols to digitally compute the noiseless (if the previous detections were correct) samples of waveform samples before $t_{k-1}$. In this manner, in the nonlinear frequency domain the noise is located only in the symbol time of the $k$-th symbol, when detecting $x_k$. The pseudocode for the DF-FNFT detection is shown in Algorithm 2.

---

**Algorithm 2** DF-FNFT detection

---

1: **Data:** Samples of the received signal $\tilde{\mathbf{r}} = (\tilde{\mathbf{r}}_1, \dots, \tilde{\mathbf{r}}_{N_b})$
2: **Result:** The detected symbols $\hat{\mathbf{x}}_1, \dots, \hat{\mathbf{x}}_{N_b}$
3: **for** $k = 1, \dots, N_b$ **do**

4:　　Apply the FNFT to $\mathbf{r}_1, \dots, \mathbf{r}_{k-1}, \tilde{\mathbf{r}}_k$
5:　　Obtain a noisy version of the nonlinear spectrum
6:　　Apply standard detection (matched filtering and sampling)
7:　　Obtain $\hat{x}_k$
8:　　If $k < N_b$, perform all the TX operations except for precompensation, to obtain $\mathbf{r}_k$, given $\hat{x}_1, \dots, \hat{x}_k$
9: **end for**

---

The computational cost of the DF-FNFT is that of one BNFT and two FNFTs. On the one hand, the FNFT evaluates the nonlinear spectrum independently on each nonlinear frequency, as for the I-FNFT, and, thus, two FNFT are used since, at the $k$-the step, the FNFT should obtain the nonlinear spectrum considering the received signal in the $k$-th time slot, and the digitally computed signal in the $k - 1$-th time slot. On the other hand, the BNFT is implemented with the Nystrom conjugate gradient method [29,30], which builds the signal samples starting from the left-most side (also considering the time inversion) and then grows on the right when considering additional symbol time—which in the DF-FNFT case are feedback symbols. As a consequence, at each step, the BNFT should be performed in a single time slot.

*4.4. The Decision Feedback-BNFT (DF-BNFT) Detection*

The decision-feedback BNFT (DF-BNFT) detection—sketched in Figure 1 in blue—uses the NFT causality property to detect symbols in the time domain, without any FNFT, differently from the I-FNFT and DF-FNFT [13]. For $k = 1, \dots, N_b$, the $k$-th symbol is selected as the one that minimizes the euclidean distance with the received signal before $t_k$, provided that the first $k - 1$ symbols have been already decided. This detection avoids the detrimental impact of the FNFT on the noise, differently from the other strategies. Furthermore, while the I-FNFT and the DF-FNFT also rely on the trick to avoid small values on $|a(\lambda)|$, the DF-BNFT does not, since the nonlinear spectrum is not evaluated at the RX. The pseudocode forDF-BNFT detection is shown in Algorithm 3.

---

**Algorithm 3** DF-BNFT detection

---

1: **Data:** Samples of the received signal $\tilde{\mathbf{r}} = (\tilde{\mathbf{r}}_1, \dots, \tilde{\mathbf{r}}_{N_b})$
2: **Result:** The detected symbols $\hat{\mathbf{x}}_1, \dots, \hat{\mathbf{x}}_{N_b}$
3: **for** $k = 1, \dots, N_b$ **do**

4:　　**for** $X_m \in \{16\text{-QAM constellation}\}$ **do**

5:　　　　Perform all the TX operations except for precompensation, to obtain to obtain $\mathbf{r}_k^{(m)}$, given

　　　　　$\hat{x}_1, \dots, \hat{x}_{k-1}, X_m$
6:　　　　Evaluate $d_m = ||\mathbf{r}_k^{(m)} - \tilde{\mathbf{r}}_k||_2$
7:　　**end for**
8:　　Obtain $\hat{x}_k = \text{argmin}_{X_m} d_m$
9: **end for**

---

As far as it concerns the computational cost, the DF-BNFT requires $M$ BNFT. Indeed, using the Nystrom conjugate gradient method [29,30], as for the DF-FNFT, at each step, the digital waveforms corresponding to 16 different symbols should be performed only in the considered time slot, since the samples on prior time instants, are taken from previous steps.

## 5. Improved Modulations

The use of the continuous spectrum for NFDM systems was originally suggested in [3,4]. The first modulation technique that has been proposed to exploit this idea is the NIS modulation, in which the QAM signal (or its standard Fourier transform) is directly mapped on the nonlinear spectrum $\rho(\lambda)$ according to Equation (1), and the optical signal is then obtained by a BNFT [4]. We refer to this mapping as "standard" modulation, which is sketched in Figure 3a. The main advantages of the NIS modulation Equation (1) are that (i) it can be seen as a natural nonlinear extension of a conventional linear modulation, to which it reduces at low power, and (ii) under certain circumstances, it can exploit the causality property reported in Section 4.1. An example of a signal encoding $N_b = 16$ symbols is shown in Figure 4a, where it is evident that the signals before and after the NFDM mapping are very similar to the first symbols, while they differ on the last symbols, where the latter loses power and acquires a tail. Both the power decay and the tail growth become more relevant as the signal power and duration increase.

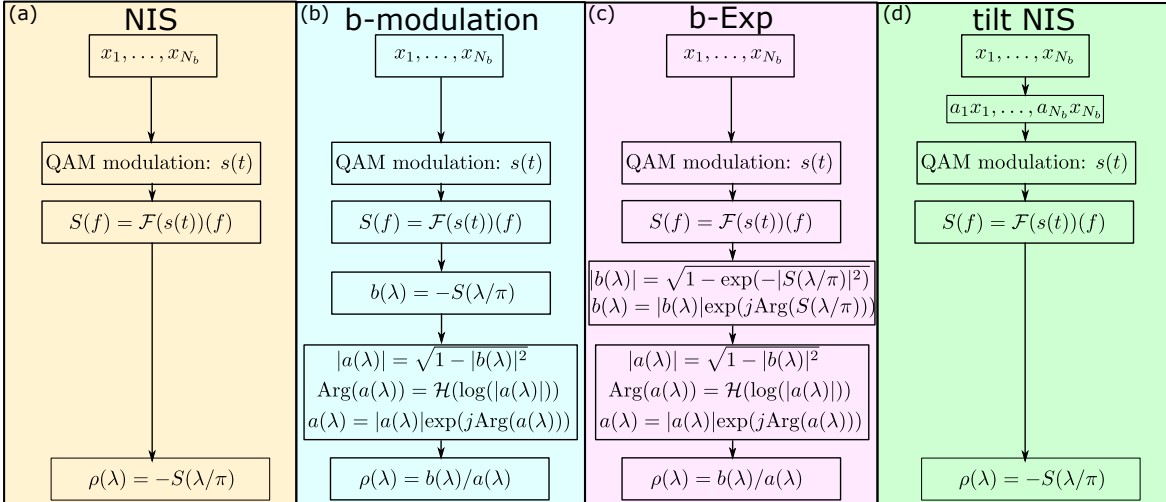

**Figure 3.** Details of modulations (**a**) NIS modulation; (**b**) b-modulation; (**c**) bExp-modulation; (**d**) NIS + tilt modulation.

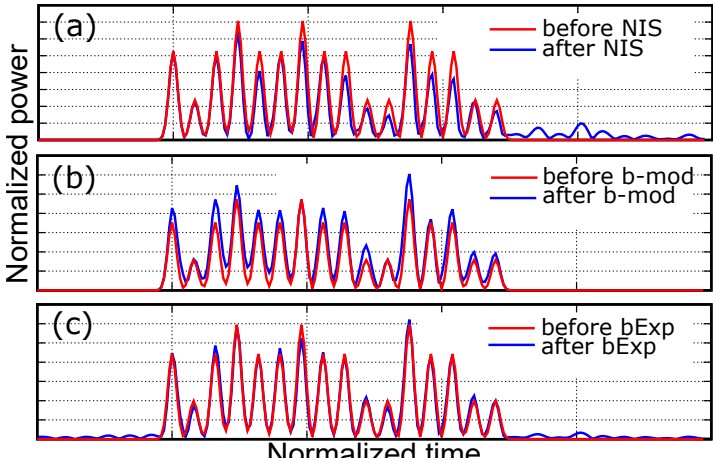

**Figure 4.** Normalized power of the quadrature amplitude modulation (QAM) signal $s(t)$ and of the corresponding optical waveform $q(t)$ for $N_b = 16$ symbols and (**a**) NIS modulation; (**b**) b-modulation; (**c**) bExp-modulation. The waveform power is 4 dBm.

On the other hand, the NIS modulation has some limitations. In particular, as shown in Section 3, it has an unfavorable behavior in terms of minimum Euclidean distance between the generated

waveforms, which is basically the reason why even the performance of the best detection strategy considered in Section 4 decays with the rate efficiency. After NIS, other techniques for modulating the continuous spectrum have been proposed [17,18,31,32]. In the following, we describe two such techniques and propose a novel power-tilt strategy to improve the NIS modulation.

### 5.1. The b-Modulation

After NIS, the b-modulation—sketched in Figure 3b—was proposed, which consists in mapping the QAM signal (or its standard Fourier transform) on the scattering data $b(\lambda)$ [31]. The continuous nonlinear spectrum $\rho(\lambda)$ is obtained (in the absence of discrete spectrum) through the following equations

$$|a(\lambda)| = \sqrt{1 - |b(\lambda)|^2} \tag{2}$$

$$a(\lambda) = |a(\lambda)| \exp\left(j\mathcal{H}\left(\log(|a(\lambda)|)\right)\right) \tag{3}$$

$$\rho(\lambda) = b(\lambda)/a(\lambda) \tag{4}$$

with $\mathcal{H}(\cdot)$ being the Hilbert transform. The main advantages of b-modulation are that (i) if the standard FNFT detection is used, the impact of noise is reduced, by avoiding the division by $a(\lambda)$ at the RX, and (ii) the temporal duration of the signal is fully controlled, as shown in Figure 4b. Furthermore, a recent theoretical study on the impact of noise in the nonlinear frequency domain (after the FNFT) showed that, differently from the standard nonlinear spectrum modulation, the power spectral density of the noise tends to zero when the energy of the signal increases [33]. Unfortunately, the power of the signal is severely limited when using b-modulation, since $|b(\lambda)| < 1$ must hold for any $\lambda \in \mathbb{R}$ [17,31]. Theoretical bounds on the power in *b*-modulated systems (and for continuous spectrum modulation in the normal dispersion regime) have been recently given [34].

### 5.2. The b-Modulation with Exponential Mapping

A slight modification of the NIS modulation, based on an exponential mapping, was proposed in [18] to allow the modulation of the continuous spectrum also in the defocusing regime (GVD parameter $\beta_2 > 0$), in which the constraint $|\rho(\lambda)| < 1$ must hold for any $\lambda \in \mathbb{R}$.

In the more practical focusing regime considered in this work ($\beta_2 > 0$), there is no such a constraint on the nonlinear spectrum, but there is an analogous constraint on the scattering coefficient $b(\lambda)$—the condition $|b(\lambda)| < 1$ mentioned above—which gives rise to the power limit issue with b-modulation. This issue can be addressed by employing an exponential mapping [32]. The resulting modulation technique—referred to as bExp in the following and sketched in Figure 3c—consists in mapping the linearly modulated signal $s(t)$ to the scattering coefficient $b(\lambda)$ according to [27,32]

$$b(\lambda) = \sqrt{1 - \exp\left(-|S(\lambda/\pi)|^2\right)} \exp\left(j\mathrm{Arg}(S(\lambda/\pi))\right), \tag{5}$$

where $S(f)$ is the conventional Fourier transform of $s(t)$. The nonlinear spectrum $\rho(\lambda)$ is then obtained by Equations (2)–(4). The issues related to the power of the signal are significantly mitigated in this case, since $|b(\lambda)| \leq 1$ by construction. This technique, as will be shown in the following, grants a significant performance improvement with respect to both NIS and b-modulation, though giving up on the control over the temporal duration of the optical signal. Figure 4c reports an example of a signal encoding $N_b = 16$ symbols with bExp-modulation, showing that the waveform acquires a tail on both sides.

### 5.3. The Power-Tilt Strategy

The power-tilt strategy is a simple and effective approach—proposed here for the first time—to improve the performance of NFDM, when NIS is employed. Based on the observations made about Figures 2b and 4a, the idea is that of decreasing the amplitude of the first symbols modulating $s(t)$

while increasing that of the last ones, trying to equalize the power along the transmitted waveform. This should cause the last symbols to have a larger impact on the waveform, while reducing the impact of the first symbols, yielding an overall improvement of the minimum distance.

This manuscript considers only a linear power tilt, characterized by the tilt strength $0 \leq \delta < 1$, but more involved tilt strategies may be investigated. Let $\bar{x}_1, \ldots, \bar{x}_{N_b}$ be the symbols to be transmitted, drawn from a regular QAM alphabet. The signal $s(t)$ is modulated by the tilted symbols $x_k = a_k \bar{x}_k$ for $k = 1, \ldots, N_b$, with

$$a_k = \sqrt{1 - \delta + 2\delta(k-1)/(N_b - 1)}. \tag{6}$$

If $\delta = 0$, no tilt is applied. It can easily be verified that the mean energy per symbol remains unchanged when $\delta$ is varied. The power-tilt strategy applied to NIS is sketched in Figure 3d. The applied tilt is then accounted for at the receiver side. The power-tilt strategy is illustrated in Figure 5a–c, by comparing the effect of a 0.9 tilt with the case where no tilt is used. Figure 5a,b show that, when applying the 0.9 tilt, the energy of the first symbols (and of the corresponding pulses of the modulated signal) is reduced, whereas that of the last ones is increased. Figure 5c reports the corresponding optical waveforms after the BNFT, showing how the tilt strategy partly compensates for the power decay of the optical signal with time.

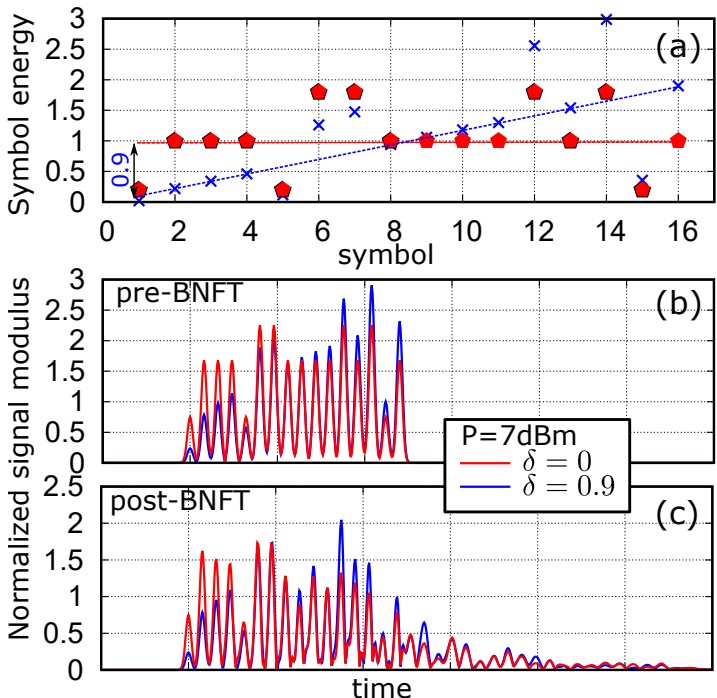

**Figure 5.** Illustration of the power-tilt strategy for a signal with 16 symbols drawn from a 16QAM constellation and a launch power $P = 7$ dBm. Blue lines and symbols refer to a power tilt $\delta = 0.9$, whereas red lines and symbols refer to the case without tilt ($\delta = 0$). (**a**) Energy of the symbols; (**b**) QAM signal $s(t)$ before the BNFT; (**c**) optical signal $q(t)$ after the backward nonlinear Fourier transform (BNFT).

Figure 6a reports the total number of bit errors accumulated after the transmission of 100 different sequences of $N_b = 1024$ symbols as a function of the symbol position in the sequence, considering the NFDM system described in Section 2 with the DF-BNFT detection, a power $P = -7$ dBm, and different tilt strengths. With no tilt ($\delta = 0$), the number of errors increases from left to right, with most of the errors occurring on the rightmost symbols. On the other hand, when increasing the tilt strength ($\delta > 0$), more energy is allocated on the last symbols and less energy on the first ones, counterbalancing the previous effect—i.e., reducing the error probability on the last symbols and increasing that on the first symbols. This improves the overall performance up to an optimal tilt value ($\delta = 0.9$ in this example).

If the power tilt is further increased, e.g., to $\delta = 0.99$, the mean energy of the first symbols becomes too low (zero, in the limit $\delta = 1$) and the corresponding error probability too high, worsening again the overall performance. The optimal tilt is the one that minimizes the average error probability, best balancing the errors occurring on the first and last symbols. Therefore, it is expected to depend on the launch power, burst length, and modulation format.

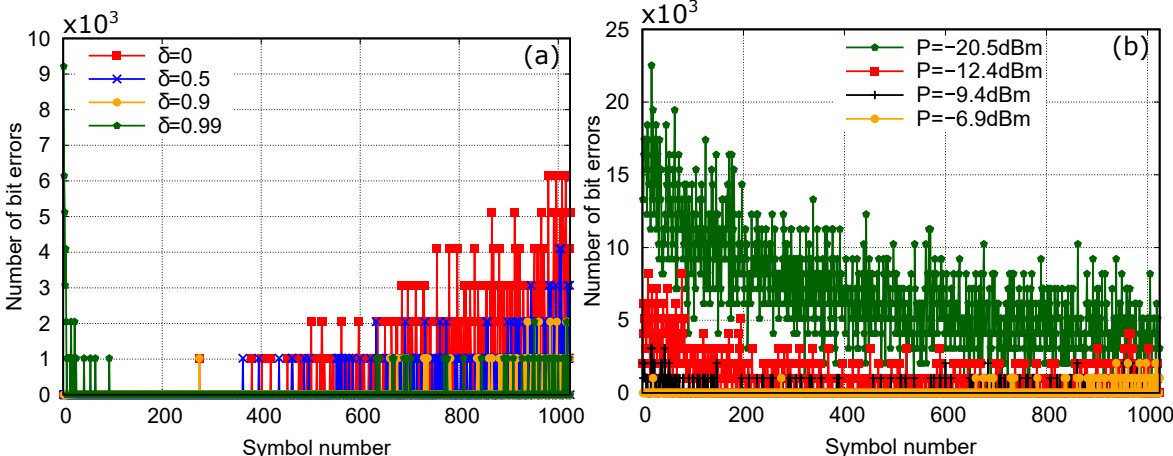

**Figure 6.** Number of bit errors versus symbol position after the transmission of 10 bursts of $N_b = 1024$ symbols (**a**) at $P = -7$ dBm for different tilt strengths, (**b**) at different power levels, for tilt strength $\delta = 0.9$.

Next, Figure 6b shows the number of bit errors accumulated in the same conditions as Figure 6a, but with a fixed tilt $\delta = 0.9$ and different launch powers. When the power is very low (linear regime) the tilt causes a lot of errors on the first symbols, as in this case the optical signal is weak and very similar to the QAM signal $s(t)$, so that the power tilt further decreases the energy of the first symbols and increases the corresponding number of errors. Conversely, when increasing the power level, the overall amount of errors decreases (as the SNR increases), and the number of bit errors per symbol starts to flatten, until that on the rightmost symbols become higher than that on the leftmost ones. This happens because, while the average signal power is high, despite the strong tilt, the signal loses energy on the latter symbols, becoming very sensitive to noise.

Finally, Figure 7a shows the performance in terms of Q-factor versus launch power obtained with the NFDM system with DF-BNFT detection and NIS modulation with different tilt strengths, for a burst length $N_b = 1024$. The figure confirms the intuition provided by Figure 6b: in the linear regime (at low power), the best performance is obtained with a small or zero tilt, while a strong tilt worsens the performance. On the other hand, at higher power, a strong tilt improves the performance. Remarkably, all the tilt strengths considered in Figure 7a improve the peak performance with respect to the case without tilt, with a maximum gain of about 2.7 dB obtained for the 0.9 tilt.

The proposed power-tilt strategy is a very simple approach to diminish the detrimental impact of the BNFT operation on the latter information symbols, with a negligible computation cost. As a possible side effect, the power tilt might increase the peak-to-average power ratio (PAPR) of the modulated optical signal, with a subsequent increase of the optical power wasted in the modulation process and of the digital resolution required at the transmitter. Indeed, inducing a power tilt on a linearly modulated systems would clearly increase the PAPR. On the other hand, the modulated signal after the BNFT is already affected by a sort of "intrinsic power tilt", which makes the signal power decay from left to right. In this case, the power tilt strategy balances this intrinsic power tilt, possibly reducing the final PAPR. In fact, we have verified by numerical simulations that, compared to the case with no tilt, a tilt of $\delta = 0.9$ slightly increases the PAPR of the optical signal at low power (where the modulation is practically linear and the tilt is actually detrimental), but slightly reduces it around the optimal power (where the tilt is indeed useful).

In principle, the power-tilt strategy can be combined with any of the detection strategies described in the previous section. However, we did not obtain any advantage when combining it with the standard FNFT detection, probably because the performance in this case is limited by the detrimental effect of the FNFT on the noise. On the other hand, the application of a power tilt to different modulation techniques—e.g., b-modulation or bExp—should be properly investigated and tailored.

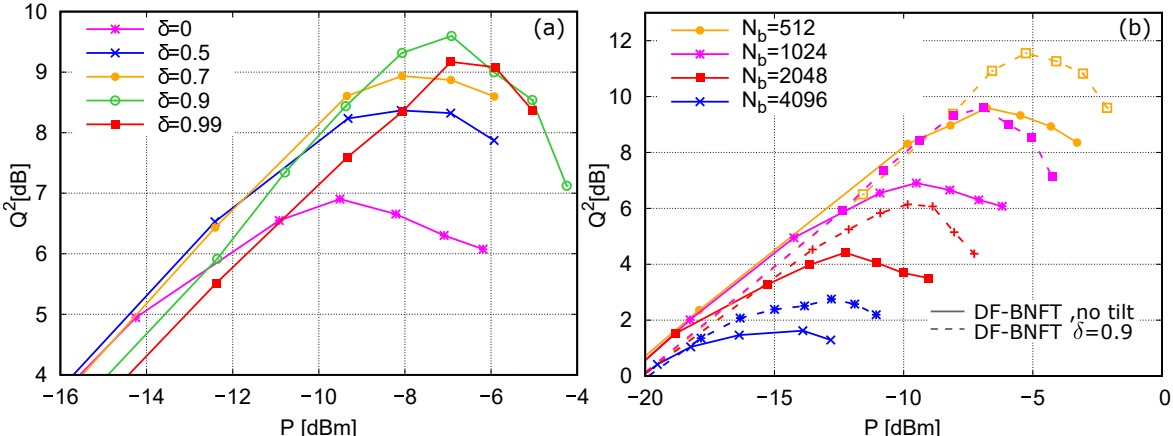

**Figure 7.** (**a**)Performance of NFDM systems with DF-BNFT detection and NIS modulation, for $N_b = 1024$ and different tilt strength $\delta$, (**b**) Performance of NFDM systems with DF-BNFT detection and NIS modulation without power tilt (solid lines) and with 0.9 tilt (dashed lines).

## 6. System Performance

First, we further investigate the performance gain that can be achieved by using the proposed power-tilt strategy. Figure 7b shows the NFDM performance as a function of the launch power $P$, comparing the cases with and without the power-tilt strategy ($\delta = 0.9$ and $\delta = 0$, respectively) and considering different burst lengths $N_b$. In both cases, the DF-BNFT detection strategy is employed, which provides the best performance among all the considered strategies. At low power, when the NFDM system tends to a conventional linearly modulated system, the performance is independent of $N_b$. On the contrary, when the power increases, the curves increase up to a certain peak value, which depends on the burst length $N_b$—the shorter $N_b$, the better the peak performance. This behavior, already shown for the standard NIS modulation [9,13,24] and discussed in Section 3, is maintained also when the power-tilt strategy (dashed line) is used. Nevertheless, the tilt strategy with 0.9 strength significantly improves the peak performance of NFDM-NIS with DF-BNFT compared to the case of a flat power allocation (solid lines), with a larger gain when the burst length is shorter. In this case, since the same tilt strength $\delta = 0.9$ is used for any launch power, the performance at low power worsens, as explained in Section 5. This undesired effect could be easily avoided by optimizing the tilt strength for each launch power. Moreover, we expect that further improvements can be obtained by optimizing the tilt strength and/or tailoring the applied power profile for each burst length. This optimization is left for a future study.

Then, we present a summary comparison of the performance achievable with the various modulation and detection techniques considered in this work, comparing it with the performance achievable by a standard NFDM system (with NIS modulation and FNFT detection) and by a conventional linearly modulated system (with no NFT and employing electronic dispersion compensation (EDC)). Figure 8 shows the peak performance (obtained at the corresponding optimal launch power) of the different systems as a function of the rate efficiency $\eta$. The proposed I-FNFT, DF-FNFT, and DF-BNFT strategies (in order of increasing complexity and performance) significantly improve the performance of the NIS-modulated NFDM system compared to the FNFT strategy, the DF-BNFT one providing a gain of about 6 dB for $\eta \geq 20\%$. Nonetheless, even with the DF-BNFT strategy and differently from the linearly modulated system, the NFDM performance keeps decreasing

with the increase of the rate efficiency $\eta$. As shown in Section 3, this is an inherent limitation of the NIS modulation—in which the increase of the power (beyond a certain limit) or of the signal duration cause a decrease of the minimum Euclidean distance between the generated waveforms—which cannot be solved by improving the detection strategy.

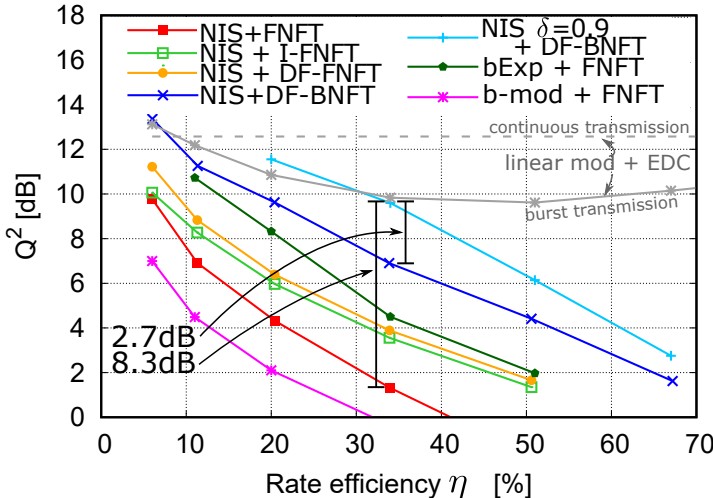

**Figure 8.** Peak performance of NFDM systems with different modulation and detection strategies, compared to a linearly modulated system with burst (solid) or continuous (dashed) transmission.

The only way in which the problem with the minimum Euclidean distance might be overcome is by modifying the modulation itself, that is, the rule with which the modulated signal is mapped to the nonlinear spectrum—for instance, by resorting to the different strategies discussed in Section 5. The b-modulation—implemented here simply by letting $b(\lambda) = -S(\lambda/\pi)$—performs worse than NIS; in this case, the peak performance is determined by the power limit discussed in Section 5.1, which is reached before a possible decay of the minimum Euclidean distance takes place; however, a proper shaping of the carrier pulses and of the modulation alphabet can increase the power limit and enhance its performance [17]. The bExp-modulation performs better than NIS with standard FNFT, I-FNFT, and DF-FNFT detections, but worse than NIS with DF-BNFT detection. Unfortunately, the DF-BNFT detection strategy is specifically tailored for the NIS modulation and cannot be directly applied to the bExp-modulation, for which the causality property in Section 4.1 does not hold.

Eventually, the power-tilt strategy proposed in Section 5.3 can further improve the performance of NIS with DF-BNFT, with a gain of 2.7 dB at rate efficiency $\eta = 34\%$, and an overall gain of almost 8.3 dB over the standard NFDM system (NIS+FNFT). Remarkably, the NFDM system with DF-BNFT detection and NIS modulation with power tilt outperforms the linearly-modulated system with EDC for rate efficiency less than approximately 30%. However, even the performance of this system—the best among all the NFDM systems considered in this work—eventually decays for higher rate efficiencies, becoming non-competitive with a conventional linearly modulated system.

## 7. Discussion

All the NFDM systems proposed so far have an important limitation: their performance decays when the signaling time is increased—the latter being a necessary condition to achieve a reasonable spectral efficiency in the presence of the guard time required by the NFT boundary conditions. This happens even in the ideal scenario considered in this work and, in fact, even over a simple AWGN channel. Thus, before delving into more realistic and detailed scenarios, it is fundamental to solve this critical issue.

In this work, we have proposed and studied several different modulation and detection strategies for NFDM systems, which can greatly improve their performance and alleviate the above-mentioned problem—unfortunately, without fully solving it. In the following, we briefly touch upon some

important aspects which have not been considered in our investigation, discussing their potential impact and their relevance in future studies on NFDM.

### 7.1. Discrete Spectrum Modulation

This manuscript considers NFDM systems in which only the continuous nonlinear spectrum is modulated: the discrete nonlinear spectrum is left empty by construction when the signal is generated at the TX, and is assumed to be still empty at the RX—if some discrete components arise during propagation because of ASE noise, they are simply neglected. Nevertheless, also the discrete spectrum can be employed to encode information, either on the eigenvalues $\lambda_i \in \mathbb{C}^+$ or, most commonly, on their components $\rho_i = b(\lambda_i)/a'(\lambda_i)$ or $b(\lambda_i)$ [3,14,16]. The modulation of the discrete spectrum can be either associated with the modulation of the continuous spectrum or employed alone.

In principle, adding some discrete components for information transmission comes for free, i.e., it should have no impact on the continuous spectrum and simply increase the information rate. Unfortunately, introducing discrete components has some drawbacks. Indeed, just a single discrete component brings a lot of energy (proportional to the imaginary part of $\lambda_i$), which strongly affects the accuracy of the numerical algorithms, so that the numerical methods usually employed for the BNFT cannot be used directly but should be combined with the Darboux transform [16,35]. Furthermore, the information carried by a few discrete components is negligible with respect to that carried by the continuous nonlinear spectrum. To the best of our knowledge, the highest number of modulated discrete components demonstrated experimentally is seven without continuous spectrum and in single polarization [22], and two combined with continuous spectrum and two polarizations [16]. Similarly to the continuous spectrum, the b-modulation has been shown to provide significant advantage with respect to the conventional $\rho_i$-modulation, also for the discrete spectrum, by avoiding the noise on $a'(\lambda_i)$. As far as it concerns the detection strategy, the conventional FNFT has been mostly used, except for the MAP strategy in time employed in [22], and a detection based on machine learning [25]. An NFT domain equalizer for discrete spectrum NFDM has been proposed in [36].

An interesting but still unexplored possibility is that of using the discrete spectrum, jointly with the continuous one, not to carry additional information, but to control the Euclidean distance between the generated waveforms, trying to solve the noise problem discussed in Section 3. Indeed, though it is theoretically possible to create a large set of high-energy waveforms with an empty discrete spectrum—for instance by using NIS modulation—there is no guarantee that these waveforms will have a large enough minimum Euclidean distance. In fact, the conventional linear modulations that are commonly used in optical fiber systems—which guarantee the desired behavior in terms of minimum Euclidean distance—give rise to optical waveforms that, at typical launch powers, do contain discrete spectral components (solitons) with a very high probability [37].

### 7.2. Dual Polarization NFDM

For the sake of simplicity, this manuscript considers only single-polarization NFDM systems. However—taking advantage of the integrability of the Manakov equation [38]—dual-polarization systems employing NIS or b-modulations have been considered in many works, reporting similar performance with respect to their single-polarization counterpart, but doubling the information rate [10,15,16,39,40]. In fact, we expect that the power-tilt strategy and the improved detection-strategies described in Section 4 can be applied to dual-polarization NFDM with little difference (but a higher complexity), nearly doubling the information rate and obtaining an overall behavior similar to that shown in Figure 8.

### 7.3. Impact of Attenuation

In this manuscript, ideal distributed amplification has been considered. Under this assumption, and in the absence of noise, the optical channel is integrable through the NFT, and the theory underlying the NFDM concept is exact. The reason for this choice is that, even in such an ideal scenario,

NFDM has not been demonstrated to outperform conventional systems. Therefore, we believe that the optimization of NFDM should be carried out in the simplest setup, and eventually tested in the more realistic one, in which fiber attenuation breaks the integrability and worsens the system performance (even when considering the lossless path-averaged model [41]).

### 7.4. Network Scenario

Ultimately, the most interesting scenario for NFDM systems is a network scenario, in which different users may share the same fiber links using different portions of the nonlinear spectrum, hence avoiding interference [18]. This is the scenario which can effectively uncover the potential advantages of NFDM over conventional WDM techniques, which cannot avoid nonlinear interference between different unknown channels. In this perspective, it is not really required that NFDM systems actually outperform linearly-modulated systems in the single-user single-channel scenario considered in this work. However, it is at least required that they perform reasonably well in this simple scenario, achieving a spectral efficiency that is comparable to that of conventional linearly modulated systems. Moreover, the actual replacement of WDM networks with NFDM networks would probably require the development of optical add-drop multiplexers that operate directly in the nonlinear spectrum domain. This is a fascinating subject to investigate but, at the moment, it remains purely at a theoretical level.

### 7.5. Computational Complexity

In recent years, many advances have been made in the development of efficient FNFT and BNFT algorithms (see, e.g., [5] and references therein). However, we are still a long way from the computational simplicity and stability of algorithms such as the FFT. Moreover, some techniques discussed in this work, such as the DF-BNFT in Section 4.4, use the FNFT or BNFT algorithms several times per each block of received symbols, further increasing the computational complexity. This is another important aspect that needs to be carefully investigated, with the aim of reducing the complexity of the NFT algorithms and of the related NFDM modulation and detection technique. However, such a complex issue is of secondary importance compared to the performance issue that is discussed above and which, according to our findings, is not yet solved.

## 8. Conclusions

In this manuscript, we have first considered a standard NFDM system, discussing its weakness with respect to conventional linearly modulated systems. In particular, we have highlighted that both the modulation and the detection strategies have been directly inherited from conventional systems and simply reapplied in the nonlinear spectrum domain, with no check on their optimality in this new context. In fact, we have shown that the linear modulation of the continuous nonlinear spectrum yields a set of optical waveforms with a minimum distance that decays as the power increases, making the system very sensitive to noise. At the same time, the detection is optimized for an AWGN channel, but is applied in the nonlinear frequency domain, where the noise is signal-dependent. Both these issues are responsible for the severe performance limitations that affect NFDM systems and which prevent them from achieving high spectral efficiencies.

In light of the above, we have investigated some improved modulation and detection strategies that can mitigate the detrimental impact of amplifier noise in NFDM systems. In particular, by combining the NIS modulation with a simple power-tilt strategy and the DF-BNFT detection strategy, we have obtained a performance improvement of more than 8 dB with respect to a standard NFDM system. The improved NFDM system outperforms a conventional linearly modulated system with EDC for a rate efficiency—the ratio between the effective signaling time and the total (signaling plus guard) time in which the channel is used—$\eta \leq 30\%$. Unfortunately, when the rate efficiency is increased, the performance decays even for the improved system—exactly as for all NFDM systems and in contrast to linearly modulated systems. While we discussed other important aspects related to NFDM that should be properly accounted for before NFDM systems can be considered for practical

applications, e.g., the modulation of the discrete spectrum or the impact of attenuation, in our opinion, the most important theoretical issue that needs to be solved is the unfavorable performance decay that characterize the current NFDM system. Indeed, the great variability of performance among the various strategies considered in this work suggests that we are still far from optimality, and that further improvements might be achievable, for instance by more finely tuning the proposed power-tilt strategy and optimizing the applied power profile beyond a simple linear tilt.

A natural question that arises from the results presented in this work is whether we should keep investigating NFDM systems and, if so, in which direction. As we have seen, current NFDM systems have some fundamental limitations which make them not competitive with traditional systems—even in an ideal physical scenario and even considering the large gains provided by the improved modulation and detection strategies proposed in this work—unless one is interested in the transmission of very short bursts separated by long guard intervals. Nevertheless, we believe that a more appropriate mathematical treatment of the noise impact on NFDM, possibly favored by a joint exploitation of the continuous and discrete spectrum, could help to design optimal modulation and demodulation techniques to eventually overcome these limitations. Without such a major breakthrough, it is unlikely that we will ever be able to exploit the alleged tolerance of NFDM to fiber nonlinearity. This is, therefore, the main research direction that we recommend for future studies on NFDM.

**Author Contributions:** Conceptualization, S.C. and M.S.; methodology, S.C. and M.S.; software, S.C.; investigation, S.C. and M.S.; resources, M.S. and E.F.; writing—original draft preparation, S.C.; writing—review and editing, M.S. and E.F. ; supervision, M.S.; funding acquisition, M.S and E.F. All authors have read and agreed to the published version of the manuscript.

**Funding:** This research received no external funding.

**Conflicts of Interest:** The authors declare no conflict of interest.

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
