# Peer review of "Mitigating the Impact of Noise on Nonlinear Frequency Division Multiplexingâ€"

_applsci, doi:10.3390/app10249099_

Round 1
Reviewer 1 Report
None.
Author Response
We are grateful to the reviewer.
Reviewer 2 Report
The authors present a review about nonlinear techniques for optical fiber transmission. In general terms the work is well presented and discussed. However there are some concerns that should be addressed.
- There are some minor typos such as "beahviour" in line 140. A careful check should be carried out.
- I strongly recommend the addition of block diagrams for illustrating both modulation and detection schemes. Although the authors have included some algorithms, the readers will thank some graphical help in this regard
- Does NFT have a power consumption penalty with respect to the linear modulation? Power consumption should be mentioned and related to power tilt.
- Why does Q^2 behave non-monotically respect to power-tilt? Is the optimal tilt dependent on the subjacent modulation (M-QAM)?
- The authors highlight that nonlinear techniques are currently outperformed by traditional modulation schemes. Why should we keep investigating on NFT-based techniques? Will these techniques improve their performance with appropriate mathematical treatment or under certain physical channel conditions? A more in-depth discussion in this regard is missing.
- Currently, networking with frequency multiplexing can be obtained using OFDMA, e.g. Why moving to a more complex scheme based on nonlinear spectra? Does it offer any other advantage although only theoretical?
- The conclusions should be improved integrating all the presented and discussed elements along the paper.
Reviewer 3 Report
The authors present an interesting summary and comparison of the potential approaches to mitigate the impact of noise in nonlinear frequency division multiplexing schemes. The introduction and references are adequate, and the results and conclusion are well presented and of interest to the field, making it a useful article for those working on the topic.
I think the manuscript can be mostly published as is. I recommend fixing a minor issue, though. Reference [1] only appears cited for the first time after reference [26], I imagine due to revisions after the first draft.
Author Response
We thank the reviewer for his work. We have corrected the mistake highlited by the reviewer.